# Perspectives of providing magnesium sulfate to patients with preeclampsia and eclampsia: A qualitative study amongst nurse-midwives in Dar es Salaam, Tanzania

**Victor Z. Chikwala**[1]*, **Agnes F. Massae**[1], **Stella E. Mushy**[1], **Edith A. M. Tarimo**[2]

**1** Department of Community Health Nursing, School of Nursing, Muhimbili University of Health and Allied Sciences, Dar es Salaam, Tanzania, **2** Department of Nursing Management, School of Nursing, Muhimbili University of Health and Allied Sciences, Dar es Salaam, Tanzania

* victorchikwala2@gmail.com

## Abstract

### Background

Preeclampsia and eclampsia are among the leading direct causes of maternal death and morbidity worldwide. Up to 34% of maternal deaths in Tanzania are due to preeclampsia/ eclampsia. Magnesium sulfate is recommended for preventing and treating convulsions in women with Preeclampsia or eclampsia. However, evidence suggests limited knowledge of its dosage and proper toxicity assessment after administration among health care providers.

### Aim of the study

This study explored nurse-midwives' perspectives on providing $MgSO_4$ to patients with pre-eclampsia or eclampsia in Tanzania.

### Materials and methods

A descriptive exploratory qualitative study using in-depth interviews was conducted to understand nurse-midwives' perspectives on providing magnesium sulfate to patients with PE/E. Nineteen nurse-midwives were interviewed from three hospitals in the Dar es Salaam region. We used a semi-structured interview guide in Kiswahili language to collect data. All interviews were digitally recorded and transcribed verbatim. We analyzed data using inductive content analysis.

### Results

This study revealed that nurse-midwives provide magnesium sulfate to save the lives of women and their unborn children. Nurse-midwives reasoned that confidence in their skill enhances provision of magnesium sulfate. However, they were concerned about its effect on the progress of labour. Ineffective use of magnesium sulfate emerged from inadequate training, an unsupportive work environment, and underutilization of the existing guidelines.

**Data Availability Statement:** All relevant data are within the manuscript and its Supporting Information files.

**Funding:** The author(s) received no specific funding for this work.

**Competing interests:** I have read the journal's policy and the authors of this manuscript have the following competing interests: All authors have no competing interests.

## Conclusion

Nurse-midwives have clear drive to provide magnesium sulfate to women with preeclampsia or eclampsia. However, inadequate training, underutilization of guidelines and unsupportive work environment lead to ineffective use of magnesium sulfate. Targeted practical training should be emphasized for nurse-midwives mastery of clinical competencies.

## Introduction

Preeclampsia and eclampsia (PE/E) are among the most common causes of maternal death and morbidities. Preeclampsia alone is responsible for more than 500,000 fetal and more than 70,000 maternal deaths each year worldwide [1], most of which occur in lower- and middle-income countries (LMIC). Up to 34% of maternal deaths in Tanzania are due to preeclampsia/eclampsia [2]. These conditions are also associated with poor fetal and neonatal outcomes, including preterm birth, low birth weight, and perinatal death [3, 4]. Early detection and appropriate treatment of preeclampsia can save lives; however, gaps exist.

Women having preeclampsia with severe hypertension or hypertension with neurological signs or symptoms are to be given anticonvulsants for seizure prophylaxis [1]. Evidence suggests that magnesium sulfate is superior to diazepam or phenytoin for prevention of convulsions among women with preeclampsia [5]. In Tanzania, Magnesium sulfate is among the essential drugs authorized to be used at all levels of health facilities [6]. However, administration of any parenteral anticonvulsants for women with preeclampsia/eclampsia is low (13.35%), with injectable diazepam (55%) being often available compared to magnesium sulfate (40.77%) [7].

The majority of nurse-midwives (90%) in Tanzania recognize magnesium sulfate as a recommended drug for seizure prophylaxis during pregnancy. However only 38.4% of them know its dosage and correct assessment of toxicity after administration [8]. To the best of our review, there is limited literature on the nurse-midwives' perspectives on providing magnesium sulfate to patients with preeclampsia/eclampsia. With the knowledge that nurse-midwives contribute to most of health workforce in Tanzania, understanding their views on providing magnesium sulfate is critical. Therefore, this study aimed to explore nurse-midwives' perspectives on the provision of magnesium sulfate to women with pre-eclampsia/eclampsia in Tanzania.

## Materials and methods

### Study design

A qualitative descriptive exploratory design was employed to understand the perspectives of nurse-midwives on administering magnesium sulfate to patients with preeclampsia and eclampsia. This design was chosen for its ability to facilitate an in-depth exploration of crucial healthcare questions, yielding insights with direct implications and significant impact on the specific healthcare setting [9]. This method was appropriate for exploring participants' opinions, beliefs, and professional judgments shaped by their personal experiences and clinical practice regarding the provision of magnesium sulfate to patients with preeclampsia and eclampsia in selected health facilities in Dar es salaam, Tanzania.

## Setting

The study was conducted in Dar es Salaam region. This region is located on the eastern coast of the Indian Ocean of Tanzania. The region has an area of 1393 square kilometers and comprises five municipals; Ilala, Kigamboni, Kinondoni, Ubungo and Temeke.

The study was conducted in two public regional referral hospitals and one private hospital located in Ilala and Temeke municipals. These municipals are densely populated with heterogeneous communities. Each regional referral hospital has nurse-midwives between 199 and 305, the maternity units have around 23 qualified nurse-midwives working in obstetrics and gynecology wards, antenatal and post-natal wards [10]. The selected hospitals provide comprehensive obstetric and neonatal emergency services and serve as referral centers for the lower-level health facilities. Also, these hospitals have authorized nurse-midwives to offer comprehensive obstetric emergency services, including the administration of essential medicines like magnesium sulfate.

## Participants and recruitment

Purposive sampling was used to recruit registered nurse-midwives working in the maternity unit after obtaining permission from administrative personnel. We recruited participants with different years of working experience to get information from multiple perspectives. Those recruited had a diploma in nursing and midwifery or a higher level of education, provided direct care, and had experience giving magnesium sulfate to a patient with PE or E. The researchers obtained a list of registered nurse-midwives working at the maternity unit from the respective nurse- midwife in-charges. We then approached each informant individually, informed them of the study's purpose and set an interview appointment. Informants away from workplace were contacted through mobile phone to arrange an appointment for an interview. Every nurse-midwife who was contacted consented in writing to take part.

## Data collection tools

This study employed a semi-structured interview guide consisting of open-ended questions, allowing participants to freely express their perspectives on administering magnesium sulfate to patients with preeclampsia and eclampsia. The researchers developed the interview guide based on insights from a thorough literature review to ensure the questions effectively addressed the research objectives. The primary research questions were as follows:

1. What are nurse-midwives' perceived competencies in providing magnesium sulfate to patients with PE/E?

2. How health facility factors influence the nurse-midwives in the provision of magnesium sulfate for women with PE/E

Questions included probes to ensure relevance of the information obtained from participants. The interview guide was initially developed in English and later translated into Kiswahili, the native language spoken by most participants, to ensure they clearly understood the questions. To ensure clarity and relevance, the interview guide was first piloted at one health facility before conducting the subsequent interviews. Moreover, sociodemographic information was collected from participants using a background information form, which included details such as age, sex, educational background, academic rank, and work experience.

                                                                                                    

## Data collection procedures

In-depth, face-to-face interviews were conducted in three hospitals from 24[th] April to 30[th] June 2022. For convenience, the data collection sites were labeled as C1, C2 and C3. The schedule for data collection was predetermined by appointment set with at each of the designated centers. The interviews were conducted at the informants' usual workplaces in a quiet room to ensure privacy, and the conversations were audio-recorded. Before the interview, all participants completed a background information form and provided written consent for both participating in and recording the interview. Only the researcher and the informant were present in the room during interview. The interviews were conducted by VZC, a male master's student in Midwifery and Women's Health. He had participated in training on qualitative research methods and has experience in qualitative data collection and analysis.

All interviews were conducted in Kiswahili because both the interviewer and the informants were native speakers. Probes were used in between main interview questions to elicit detailed information and guarantee that participants' answers were clear. Interviews went on until information saturation was reached, at which point participants recruitment was concluded [11]. The interviews lasted between 30 and 60 minutes.

## Data analysis

Qualitative content analysis of this study was guided by Graneheim and Lundman [12], to understand the transcripts. All interviews were transcribed verbatim. VZC and an Intern Nurse transcribed the audios and all authors were involved in data analysis. The transcripts were read repeatedly to search for the meaning and deeper understanding. The condensed meaning units were formed and coded. Coding was done manually by placing labels alongside the margins of the transcripts. VZC developed initial codes which were then discussed and reviewed by AFM (BScN, MSc, PhD), SEM (BScN, MSc, PhD), and EAMT (BScN, M.Phil, PhD). All agreed codes were exported to Microsoft excel and sorted into subcategories based on their similarities, and the latter were grouped into categories. All categories were constructed in Kiswahili to avoid losing the original meaning. The categories were then translated from Kiswahili to English for reporting.

## Ethical considerations

The study was conducted in accordance with relevant guidelines and regulations. Ethical approval was granted by the Senate Research and Publication Committee, the institutional review board of the Directorate of Research and Publications at Muhimbili University of Health and Allied Sciences (Ref. No. MUHAS-REC-03-2022-1038). Permission to collect data was obtained from the health facility authorities. Written informed consent was obtained from each participant prior to the commencement of interviews. The signed consent form outlined participants' rights to participate in and withdraw from the study, its risks and benefits, and the audio recording of the interview. Throughout the investigation, the researcher protected confidentiality and treated the informants' identities in an anonymous manner.

## Results

Nineteen participants were included in the study. The participants' age ranged from 26 to 57 years old. The majority (89.5%) of the participants were females and most (94.7%) of them had a diploma education level. More than half (52.6%) of the participants had more than five years of experience in taking care of PE/E patients while 26.3% had 3–5 years of experience (Table 1).

**Table 1. Demographic characteristics of study participants (n = 19).**

| Characteristic | Frequency (n) | Percent (%) |
|---|---|---|
| **Age group (Years)** | | |
| 18–35 | 9 | 47.4 |
| 36–55 | 9 | 47.4 |
| >56 | 1 | 5.2 |
| **Sex** | | |
| Male | 2 | 10.5 |
| Female | 17 | 89.5 |
| **Level of education** | | |
| Diploma | 18 | 94.7 |
| Degree | 1 | 5.3 |
| **Working experience (Years)** | | |
| < 3 | 4 | 21.1 |
| 3–5 | 5 | 26.3 |
| > 5 | 10 | 52.6 |

This study explored nurse-midwives' perspectives on the provision of magnesium sulfate to patient with PE/E. From the data analysis three categories and their subcategories emerged and are discussed in details below (Table 2).

## a) Knowledge of magnesium sulfate provision

This category describes the process, participant perceptions and competencies for administering magnesium sulfate. Nurse-midwives had current knowledge on provision of magnesium sulfate, they were willing and confident to provide magnesium sulfate however gaps are present. Four subcategories have been described under this category; Indications for the use of magnesium sulfate, determining the dosage regimen, monitoring the patient's response, and postponing the administration of magnesium sulfate.

**Indications for magnesium sulfate use.** Participants reported correct indications for providing magnesium sulfate. First, they identified the right patients and determined the dosing regimen. Participants used magnesium sulfate to prevent seizures in women with severe preeclampsia features such as high blood pressure, urine protein greater than +2 on deep stick, severe headache, upper abdominal pain, and blurred vision. They also used it to treat convulsions in women who have eclampsia.

**Table 2. Emerging categories and subcategories on nurse-midwives' perspectives of magnesium sulfate provision.**

| Category | Subcategory |
|---|---|
| a) Knowledge of magnesium sulfate provision | Indications for magnesium sulfate use |
| | Determination of dosage regimen |
| | Monitoring of patient response |
| | Postponement of magnesium sulfate administration |
| b) Reasons for magnesium sulfate use | Perceived benefits of using magnesium sulfate |
| | Confidence with own skill to give magnesium sulfate |
| c) Barriers to magnesium sulfate provision | Individual barriers to magnesium sulfate provision |
| | Institutional barriers to magnesium sulfate provision |

"*For eclamptic with a gestation of 20 weeks and more . . . I will give magnesium. For others with preeclampsia who have severe features like epigastric pain, headache, double vision, epigastric pain and other things, BP [systolic] 140 and higher and diastolic 90, urinary protein plus 3 and these severe features we give magnesium sulfate*"

(Participant 11).

"*We often check if the patient has convulsed, if there is protein in the urine starting from plus two or three, and also if there are any severe features, such as epigastric pain, headache, and blurred vision. These are the indicators we look for to administer Magnesium sulfate*"

(Participant 10).

**Determination of dosage regimen.** Participants stated that they used two dosage forms namely; loading and maintenance doses. They further explained that the loading dose consisted of 14 g magnesium sulfate divided into 4 grams intravenously and 10 grams intramuscularly.

"*For a patient who has just arrived from home and was brought in by relatives, we will initiate the loading dose, which is 14 grams. We administer 4g intravenously (IV) and 10g intramuscularly (IM)*"

(Participant 8).

For intravenous administration, participants commonly used 50% magnesium sulfate diluted to 20% by adding 12 mL of water for injection, while that for intramuscular injection was supplemented with 1 mL of 2% lignocaine to reduce pain associated with magnesium sulfate.

"*For the loading dose, we need to reduce the concentration from 50% to 20%. To achieve this, we mix 8ml of 50% magnesium sulfate with 12ml of water for injection, resulting in 20ml of 20% magnesium sulfate. We then administer 4g (20ml) slowly via IV over twenty minutes. After that, we administer 5g intramuscularly (IM). To do this, we dilute 5g to 10ml and add 1ml of 2% lignocaine before giving a deep IM injection*"

(Participant 2).

Participants further reported that after the loading dose, 5 grams was injected deeply intramuscularly every four hours, giving six maintenance doses in 24 hours. Participants emphasized that for recurrent seizures, 2 grams of magnesium sulfate was slowly administered by intravenous injection.

"*Therefore, you continue with a maintenance dose of five grams every four hours, alternating between each buttock, until six doses are administered within twenty-four hours*"

(Participant 17).

"*The mother can fit fifteen minutes or ten minutes before you go to the first maintenance dose. So, you take the 8 Mls of magnesium sulfate which you divide into two, it becomes 4mls plus 6cc water for injection it becomes 10mls, you give her IV slowly*"

(Participant 6).

**Monitoring of patient response.** Monitoring of women receiving magnesium sulfate for signs of recovery, side effects or toxicity was important. Participants reported making observations prior to administering the repeat dose within the 4 hours interval. Verification of vital signs, urine output and color, and fetal heart rate was emphasized.

"*You will check her [the woman] breathing rate, and then you can look at the knee flexibilities. You can tell her to bend her legs, if she has no coordination at all you use the patellar harmer, if you see the mother has no coordination you know that things are not going well here and even the color of the urine if it is too dark will give you some information*"

(Participant 8).

While monitoring patients who have been given magnesium sulfate, participants noted several reported effects, including intoxication, pain and burning at the injection site, drowsiness, and prolongation of labor.

"*Magnesium sulfate has several side effects that we commonly identify. The mother may experience general weakness, pain at the injection site, and headaches as a result of the medication's side effects*"

(Participant 9).

"*It happens very slowly, meaning when the mother has contractions and you inject the magnesium sulfate, the labor pain completely ceases. Once the medication starts to wear off, it's like the labor pain starts anew, from the beginning. For the majority of mothers, after the injection, they calm down as if they had no labor pain at all*"

(Participant 13).

Participants were asked how frequently they monitor patients who received magnesium sulfate and how they assess the patellar reflex for signs of toxicity. They reported monitoring blood pressure every 30 minutes initially, and then every 2 hours. However, they encountered difficulties in eliciting the patellar reflex.

"*I'll probably check the blood pressure every half hour if I see the mother is in more danger. I'll continue monitoring her until the blood pressure stabilizes, then maybe every 2 hours. I'll keep checking her and asking, 'How are you feeling?'*"

(Participant 5).

"*Performing the patellar reflex here is very rare. Personally, I only do it if I am with a doctor. I cannot perform the patellar reflex on my own*"

(Participant 16).

They also reported the possibility of bleeding complications, exposure to operative delivery, suffocation of the baby at birth, and stillbirth.

"*I believe the baby could be affected because magnesium, if administered for too long, can potentially lead to issues like a low APGAR score or even stillbirth. You see, because of this risk, they caution that exceeding the recommended time could result in these outcomes*" (Participant 1).

**Postponement of magnesium sulfate administration.** The participants reported reasons which lead to the abandonment of magnesium sulfate use. These included signs of toxicity such as respiratory depression, loss of tendon reflexes, low blood pressure and low urine output.

"*So, if a patient develops magnesium toxicity, first, their urine output will be less than 30ml per hour. Also, their patellar reflex can become hypoactive, and their respiration can be below 16. In that case of magnesium toxicity, we stop the magnesium*"

(Participant 19).

Participants explained that they delayed to give magnesium sulfate for the woman with strong uterine contractions and impending childbirth because they believed it interfered with the woman's ability to push.

"*When I administer magnesium sulfate to her, she obviously becomes drowsy, doesn't push at all. . . so she feels the pain but doesn't even cry*"

(Participant 8).

Some participants did not continue with repeat dose of magnesium sulfate when the woman's diastolic blood pressure was 90mmHg or low.

"*If I measure the blood pressure and find it below normal, such as 90 or 100, I do not administer magnesium sulfate*"

(Participant 14).

## b) Reasons for magnesium sulfate use

The intention of using magnesium sulfate to treat women with preeclampsia/eclampsia was described under two subcategories. These are: benefits of using magnesium and confidence with own skill to give magnesium sulfate.

**Benefits of using magnesium sulfate.** Participants were willing to provide magnesium sulfate because they understood its benefits and the effects of preeclampsia and eclampsia on the mother and her unborn child. Their decision to use magnesium sulfate was based on the idea that it is safe to use, and saves the life of the woman and the unborn child by reducing blood pressure, preventing the effects of convulsions, and providing fast and lasting relief.

"*It's compelling to me because magnesium sulfate provides rapid relief and maintains it for an extended period with its maintenance doses. This allows continuous monitoring and protection of the patient for at least 24 hours, unlike Valium, which wears off sooner*"

(Participant 4).

"*Injecting the mother with magnesium sulfate ensures that the baby will be born alive, benefiting both the mother and the baby's survival. This treatment is crucial for managing conditions like preeclampsia or eclampsia, ensuring a safe delivery for the mother and her baby*"

(Participant 8).

Moreover, participants pointed that magnesium sulfate has saved numerous lives of mothers and their unborn babies. Without timely administration of this drug, many would have faced the risk of loss or long-term disability.

"*Magnesium sulfate helps stop or reduce seizures so she [the woman receiving magnesium sulfate] automatically benefits because if she continues to convulse her brain will be damaged which is a serious side effect later until she has a stroke or disability*"

(Participant 1).

**Confidence with own skill to give magnesium sulfate.** Participants expressed confidence with their ability to administer magnesium sulfate. Their confidence came from the ability to cope with difficult cases, adequate preparation of emergency kit and team members, individual commitment to acquire skills, knowledge of magnesium sulfate and working experience.

"*It is because of that knowledge that you also have daily understanding, something that you practice every day is the main remedy here [working in Eclampsia ward], so it means according to that [knowledge and experience] I get confidence*"

(Participant 12)

"*I am very confident because I have provided that service several times, so I have experience in identifying patients who can be given Magnesium sulfate*"

(Participant 9).

"Actually, I usually don't have fear when administering magnesium. I know my patient is a serious case, so saying I have no fear"

(Participant 16).

### c) Barriers to magnesium sulfate provision

Albeit the awareness of beneficial aspects of magnesium sulfate, participants reported limitations for providing magnesium sulfate that were classified into individual and institutional barriers, as described in the following sections.

**Individual barriers.** Even though nurses- midwives provided magnesium sulfate to women with preeclampsia or eclampsia, they also faced individual limitations during the procedure. They reported infrequent contact with the eclamptic patients, and fear of side effects as barriers to magnesium sulfate provision. Participants emphasized that administering magnesium sulfate necessitated a high level of knowledge and careful attention.

"*You have to be extremely cautious when administering this medication. It's crucial to know exactly what you're doing and to closely monitor its effects*"

(Participant 3).

Participants observed that midwives who did not give magnesium sulfate were afraid of taking care of eclamptic patients. They also reported a lack of confidence caused by insufficient hands-on experience and fear of making mistakes with magnesium sulfate administration, as stated below.

"*I think many are afraid, they haven't given the drug for a long time, they think maybe they will make a mistake, and some say that when we start loading dose, because we don't have the tendency, we forget to give magnesium sulfate*"

(Participant 15).

Participants felt their knowledge was imperfect; they expressed the need for more training to get current updates, learn more about side effects, and identify and correct magnesium sulfate toxicity. Some were unsure of what to do because what they learned in class differed somewhat from actual practice.

"*To be honest what I learned and what I came across is something else! What you learn in college is not what you do, but there is a balance between what you studied there and seeing actual patients*"

(Participant 7).

**Institutional barriers.**   Participants reported barriers to the use of magnesium sulfate that coined from the healthcare facility. Participants expressed inadequate training opportunities and refresher courses to increase their knowledge and skills in using of magnesium sulfate. Some reported that they were missing current updates because they attended seminars and workshops long time ago. The lack of on-the-job training was more pronounced among participants.

"*I said maybe there are others who get training. I mean that in a private sector like this we do not get on the job training*"

(Participant 1).

"*If I don't receive the ABCDs of these things, I'll keep doing the same thing, even if I've heard there might be another solution that I'm not aware of*"

(Participant 8)

Underutilization of guidelines for magnesium sulfate use was also reported by participants. Senior participants were not using any guidelines because they felt they had internalized the necessary procedures and did not have to go through the guidelines. However, juniors had a different experience, most of them being tutored orally and never seeing any guidelines.

"*I only saw it once I don't know it lasted a week but when I arrived, I just asked my colleagues to teach me how to administer [Magnesium Sulfate] because I forgot what I was taught in school, they just taught me orally*"

(Participant 13).

Others stressed that the guidelines were difficult to understand and took some time to get used to. They suggested that work aids instructions on what to do when providing magnesium sulfate should be clear.

"*It requires a person to stick to the point until he understands the guideline . . ..it takes time to understand, I think we need a good alternative that directs the person*"

(Participant 2).

Lack of equipment, unsupportive infrastructure, and staff shortages limited the provision of magnesium sulfate. Participants reported lack of equipment such as monitors, infusion pumps, working and filled oxygen cylinders, suction machines and patella harmers, which were important when administering magnesium sulfate.

"*We face a shortage of machines that can continuously monitor oxygen saturation and other vital signs while patients are using magnesium sulfate*"

(Participant 4).

Administration of the magnesium sulfate loading dose requires 20ml syringes but they were in short supply. Instead, participants used two 10ml syringes.

"*But we don't have the 20cc syringes, so we use the 10cc syringes instead.*"

(Participant 5).

The participants were dissatisfied with the infrastructure of the wards and felt the necessity to set up a special ward for women with pre-eclampsia or eclampsia in all referral hospitals to facilitate be monitoring of the patients.

"*We keep eclamptic patients together with other patients but often we care for them in the labour ward and the nurse stays in the labor ward, instead there should be a special section*"

(Participant 3)

Furthermore, even in a hospital with a special department, ward space and eclampsia beds were insufficient to ensure patient safety.

"*The ward isn't sufficient; there are times when two eclampsia patients have to share one bed. Imagine if they all fit? The space is very limited, clearly not enough.*"

(Participant 13)

"*I haven't seen a sturdy bed for someone who's having convulsions. How many do we have here? Have you seen one?*"

(Participant 10)

## Discussion

This study revealed that nurse-midwives have adequate knowledge of the benefits of magnesium sulfate and its indications for use in women with preeclampsia or eclampsia. However, knowledge gaps exist. The Pritchard regimen (intravascular and intramuscular administration of magnesium sulfate) was commonly and correctly used. Confidence in administering magnesium sulfate and the belief that it can be safely used influenced nurse-midwives decision to provide it. Individual and health-related factors limited the provision of magnesium sulfate.

Nurse-midwives' knowledge of the benefits of magnesium sulfate facilitated its administration to appropriate patients. This implies that nurse-midwives can confidently provide magnesium sulfate when the benefits are known. This finding provides an important message for designing on-the-job training for nurse-midwives, focusing on the benefits of using magnesium sulfate for women with pre-eclampsia and eclampsia. Similarly, a previous study conducted in Dodoma showed that nurse-midwives' knowledge facilitated provision of magnesium sulfate [8]. The similarity of these results may be due to nature of the studied contexts as well as the training curriculum among study participants.

Nurse-midwives identified important indications for providing magnesium sulfate such as hypertension associated with severe features, proteinuria $\geq$. However, they did not mention

other signs such as end-organ damage that can necessitate its provision in the absence of hypertension according to national and international guidelines [6, 13]. The International Society for the Study of Hypertension in Pregnancy (ISSHP) recommends prioritizing the evaluation of urinary protein excretion and serum creatinine if resources are limited [14]. However, nurse-midwives did not highlight serum creatinine results as significant factors for diagnosing or deciding on the administration of magnesium sulfate. This finding may be related to lack of updates or unawareness of guidelines amongst nurse-midwives. Authors in Egypt have reported similar findings, noting that a majority of nurses face challenges with their knowledge on the early detection and proper treatment of pre-eclampsia, highlighting the need for further training and updates in this area [15].

In the current study, nurse-midwives often preferred the standard Pritchard regimen over the Zuspan regimen. This preference could be attributed to the perceived practicality of the Pritchard regimen, which does not require ICU monitoring or the use of infusion pumps. This regimen's widespread adoption involved administering magnesium sulfate both intravenously and intramuscularly. Preference for this regimen could be related to inadequate skills to monitor intravenous infusions of magnesium sulfate, poor work environment and unawareness of the alternative regimen (Zuspan). In contrast, authors in high-income countries, such as the United States, have reported that the resources required to monitor a patient on magnesium infusion were a deterrent to using magnesium sulfate [16]. Most clinicians expressed a preference for an alternative medication. In the current study, participants administered both loading and maintenance doses of magnesium sulfate to all patients with indications for its use, demonstrating adherence to the clinical protocol. In contrast, another study reported inconsistencies in providing these doses, with clinicians varying the administration based on patient conditions, sometimes resulting in underdosing [16, 17].

A knowledge gap does not only exist in magnesium sulfate administration, as noted in a Kenyan study [4]. There is often lack of knowledge on how to monitor magnesium sulfate toxicity [13]. In the current study, the patellar reflex was mentioned but explanations on how to assess the reflex and interpretation of the results of assessment was lacking. Moreover, the frequency of monitoring signs of magnesium sulfate toxicity as mentioned by participants was lacking when compared to WHO guidelines [13]. This finding has clinical implications for patient safety and highlights a potential knowledge and skill gap that should be addressed through on-the-job training for nurse-midwives.

Fear of adverse consequences caused midwives to eventually postpone or cease giving magnesium sulfate. This suggests awareness of the potential for customer toxicity. Similar results indicating concerns about fetal hazards and mother safety have been documented elsewhere [18, 19]. Thus, it is correct to stop the administration of magnesium sulfate in the presence of signs of toxicity [6]. In high-income countries, healthcare providers often express concerns about the safety profile of magnesium sulfate and believe that an alternative agent for preventing eclampsia is necessary [16]. On the other hand, Brazilian obstetricians expressed concerns about potential adverse effects, such as cardiorespiratory arrest, associated with the use of magnesium sulfate, and felt inadequately prepared to manage such complications [18].

The nurse-midwives in the current study delayed the administration of magnesium sulfate because of fear that the women will be unable to push during childbirth. This finding has not been reported elsewhere. Prolongation of labor may not be a reason to discontinue or delay administration of magnesium sulfate in women with severe features of preeclampsia or eclampsia. The risk of developing eclampsia continues throughout the birth process. This finding underscores the importance of nurses' knowledge about the course of preeclampsia and eclampsia, as well as the necessity for them to be equipped with the skills needed to manage a woman who has been given magnesium sulfate during or near childbirth.

This study found that the underutilization of magnesium sulfate was caused by a lack of equipment, a paucity of midwives, and the perception that the treatment required a lot of work. The results are consistent with a several studies that demonstrated the difficulty of administering magnesium sulfate, making observations, and documenting the process, in addition to the need for insufficient midwives and equipment, such as blood pressure monitors [16, 17, 19, 20]. The similarity of the study areas' low socioeconomic standing and severely underdeveloped healthcare systems may account for the consistency of these findings. Also, these studies revealed that nurses-midwives felt the need to set up a dedicated ward for pre-eclampsia and eclampsia patients, suggesting that separate wards are needed to improve quality of care. Inadequate training was also identified as a limiting factor for the providing magnesium sulfate at the healthcare facility level. Studies from elsewhere have shown similar findings where limited knowledge and little experience instilled fear and limited the use of magnesium sulfate [16, 20]. The change in clinical guidelines as new evidence emerge should go hand in hand with the training of nurse-midwives, who are among the key implementers of the guidelines.

## Study limitations

Purposive sampling was used to recruit participants. Thus, the results may not represent the views of all midwives. The study was based on reported perspectives of providing magnesium sulfate to women with pre-eclampsia/eclampsia. Participants may have reported the information they wanted, but not what they were doing because they were not observed. The researchers used probing to minimize this bias. In addition, experiences from private hospitals may not have been comprehensively captured as only one private hospital was involved compared to two public hospitals with essentially the same characteristics. This was due to the lack of permission to conduct the study in a private hospital during the data collection. Nevertheless, the presented findings are based on the principles of saturation.

## Conclusion

Our study has revealed that nurse-midwives possess knowledge of the indications and benefits of magnesium sulfate, along with confidence in their skills to administer it. These factors are essential for the effective provision of magnesium sulfate. We also found that nurse midwives delay administration of magnesium sulfate when the woman in labor is near childbirth. Significant gaps in knowledge and skills are evident in the monitoring and care provision for patients who have received magnesium sulfate. Moreover, nurse-midwives face various challenges ranging from lack of on job training focusing on magnesium sulfate administration and care for patients with preeclampsia and eclampsia, shortage of equipment and supplies as well as unfavorable working conditions.

Based on our findings we recommend that nurse-midwives be trained with focus on mastery of practical competencies that are coherent with clinical work to minimize the gap between theoretical knowledge and practice. Monitoring and caring for patients receiving magnesium sulfate should be a focal point of on-the-job training for nurse-midwives. Additionally, it is crucial to develop clinical observation protocols that specify the frequency of patient assessments while on magnesium sulfate and provide prompts for appropriate actions based on assessment findings. Healthcare facility authorities can utilize our findings to design interventions aimed at enhancing working conditions and mitigating facility-related barriers faced by nurse-midwives when administering magnesium sulfate to patients with preeclampsia and eclampsia. Further research is needed to establish the impact of postponing magnesium sulfate administration until the woman have given birth.

## Supporting information

**S1 Table. Analysis codebook for PE_E acknowledgments.** The authors extend their sincere gratitude to the Muhimbili University of Health and Allied Sciences for providing logistic support. Our thanks go to hospital administration for their support during data collection and study participants for participating in the study. We would also like to acknowledge Kornel Izdory Metheo for assisting with transcription and recruitment of participants.
(DOCX)

## Author Contributions

**Conceptualization:** Victor Z. Chikwala, Agnes F. Massae, Edith A. M. Tarimo.

**Formal analysis:** Victor Z. Chikwala, Agnes F. Massae, Stella E. Mushy, Edith A. M. Tarimo.

**Investigation:** Victor Z. Chikwala, Agnes F. Massae.

**Methodology:** Victor Z. Chikwala, Agnes F. Massae, Edith A. M. Tarimo.

**Supervision:** Edith A. M. Tarimo.

**Validation:** Agnes F. Massae, Stella E. Mushy.

**Writing – original draft:** Victor Z. Chikwala.

**Writing – review & editing:** Victor Z. Chikwala, Agnes F. Massae, Stella E. Mushy, Edith A. M. Tarimo.

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
