## [Decision Letter · Decision Letter 0]

27 May 2024

PONE-D-24-06856Perspectives of providing magnesium sulfate to patients with preeclampsia or eclampsia: A qualitative study amongst Nurse-midwives in Dar es Salaam, TanzaniaPLOS ONE

Dear Dr. Chikwala,

Thank you for submitting your manuscript to PLOS ONE. After careful consideration, we feel that it has merit but does not fully meet PLOS ONE’s publication criteria as it currently stands. Therefore, we invite you to submit a revised version of the manuscript that addresses the points raised during the review process.The changes we require for acceptance of this article include addressing all the reviewers' comments and suggestions (see below)Where there is any conflicts between the reviews, kindly focus your attention on the reviews that require you to revise your manuscript Kindly rework the entire manuscript, especially the result section to have more than one perspectives (quotes) for every theme/subtheme; the discussion & conclusion sections need to be more rigorous and extensive, containing policy or strategic implications of your findings.   ==============================Please submit your revised manuscript by Jul 11 2024 11:59PM. If you will need more time than this to complete your revisions, please reply to this message or contact the journal office at plosone@plos.org. Please include the following items when submitting your revised manuscript:A rebuttal letter that responds to each point raised by the academic editor and reviewer(s). You should upload this letter as a separate file labeled 'Response to Reviewers'.A marked-up copy of your manuscript that highlights changes made to the original version. You should upload this as a separate file labeled 'Revised Manuscript with Track Changes'.An unmarked version of your revised paper without tracked changes. You should upload this as a separate file labeled 'Manuscript'.

We look forward to receiving your revised manuscript.

Kind regards,

Ayodeji Babatunde Oginni, MSc

Academic Editor

PLOS ONE

Journal Requirements:

2. In the online submission form, you indicated that the local and country regulation and ethical guidelines for data sharing limit data release publicly. If some researchers are interested, the data can be available on a reasonable request to the first author and the chairperson of the senate, research, and publication committee of Muhimbili University of Health and Allied Sciences (MUHAS) at the following email: drp@muhas.ac.tz

Reviewers' comments:

Reviewer's Responses to Questions

**Comments to the Author**

1. Is the manuscript technically sound, and do the data support the conclusions?

Reviewer #1: Yes

Reviewer #2: Partly

Reviewer #3: Yes

2. Has the statistical analysis been performed appropriately and rigorously?

Reviewer #1: No

Reviewer #2: N/A

Reviewer #3: Yes

3. Have the authors made all data underlying the findings in their manuscript fully available?

Reviewer #1: Yes

Reviewer #2: No

Reviewer #3: Yes

4. Is the manuscript presented in an intelligible fashion and written in standard English?

Reviewer #1: Yes

Reviewer #2: Yes

Reviewer #3: Yes

5. Review Comments to the Author

Reviewer #1: As this study is quite old one and have been published extensively before,however, It is novel in view of directed to midwifes and nurses nevertheless, I have the following comments:

-Comment1:Abstract:mention:preeclampsia or eclampsia should:"and" as they are different entities.

_Design of the study:should mention "Questionnaire." method of research.

-Comment 3:Statistical analysis:much is needed to explain regarding the statistical methods used and the results extracted from the analysis.

-Comment4:Refrrences:As the study is quite old one and has been investigated extensively before,I wish the authors to gather and cite a more recent refrrences(2021/2022/2023 and 2024).

Reviewer #2: The manuscript presents original research. However, the explanation of the methodological design seems weak and the data collection process needs to be better explained. The manuscript still lacks a clear statement on data availability. In terms of clarity, there are some grammatical inconsistencies, but nothing serious.

Reviewer #3: Dear authors,

Congratulations on conducting this intriguing study. This presents a valuable insight into reducing maternal mortality caused by this preventable factor. I have compiled my comments and suggestions, believing they will enhance the quality of your paper once uploaded.

6. PLOS authors have the option to publish the peer review history of their article (what does this mean?). If published, this will include your full peer review and any attached files.

**Do you want your identity to be public for this peer review?** For information about this choice, including consent withdrawal, please see our Privacy Policy.

Reviewer #1: **Yes: **Mohsen M A Abdelhafez

Reviewer #2: No

Reviewer #3: No

---

## [Author Response · Author response to Decision Letter 0]

11 Jul 2024

Perspectives of providing magnesium sulfate to patients with preeclampsia and eclampsia: A qualitative study amongst Nurse-midwives in Dar es Salaam, Tanzania.

Response to reviewers’ comments

We would like to thank all the reviewers for their valuable comments, which have significantly improved our work. We have revised the entire manuscript and addressed each comment point by point, as detailed in the table below.

Editor’s Comments

1 Kindly rework the entire manuscript, especially the result section to have more than one perspective (quotes) for every theme/subtheme; the discussion & conclusion sections need to be more rigorous and extensive, containing policy or strategic implications of your findings.

 Author response: Thank you for your comment. We have reworked the entire manuscript, revised the result, discussion and conclusion sections.

2 Please include your full ethics statement in the ‘Methods’ section of your manuscript file. In your statement, please include the full name of the IRB or ethics committee who approved or waived your study, as well as whether or not you obtained informed written or verbal consent. If consent was waived for your study, please include this information in your statement as well.

 Author response: Thank you for your comment. We have added a full ethics statement in the ‘Methods’ section of our manuscript file. We obtained written consent from participants and this information has been included as well.

Reviewers’ Comments

1 Is the manuscript technically sound, and do the data support the conclusions?

Reviewer #2: Partly Author response: We appreciate your comment, we have revised the conclusion accordingly 

2 Has the statistical analysis been performed appropriately and rigorously?

Reviewer #1: No Author response: Thank you for your comment. However, we could not perform statistical analysis because this was a qualitative study.

3.Have the authors made all data underlying the findings in their manuscript fully available?

Reviewer #2: No Author response: We accept your comment, we have uploaded our codebook as supplementary information.

4 Is the manuscript presented in an intelligible fashion and written in standard English?

All reviewers: Yes, Author response: Thank you for the comment

Reviewer No 1

Comment 1 Abstract: mention: preeclampsia or eclampsia should: "and" as they are different entities.

 Author response: Thank you for your comment, pre-eclampsia or eclampsia has been changed to pre-eclampsia and eclampsia. 

Comment 2 Design of the study: should mention "Questionnaire." method of research. 

 Author response: We appreciate your comment. However, this study employed an exploratory qualitative design and utilized in-depth interviews to gather qualitative data. An interview guide (questionnaire) with open-ended questions was used, as detailed in the Data Collection Methods and Guides section. 

Comment 3 Statistical analysis: much is needed to explain regarding the statistical methods used and the results extracted from the analysis.

 Author response: Thank you for your comment. As this was a qualitative study, no statistical tests could be performed. Instead, the textual data were analyzed using qualitative content analysis. However, we have conducted descriptive statistical analysis specifically on the demographic characteristics of the participants. 

Comment 4 References: As the study is quite old one and has been investigated extensively before, I wish the authors to gather and cite a more recent references (2021/2022/2023 and 2024). 

 Author response: Thank you for your comment. We have updated our references to include more recent sources. However, we retained a few references from 2017, 2018, and 2019 because we could not find most relevant studies to replace them

Reviewer #2:

Comment 1 The manuscript presents original research. However, the explanation of the methodological design seems weak and the data collection process needs to be better explained. 

Author response: Thank you for your comment we have updated the justification to the chosen design to improve the explanation on methodological design. The data collection process has been split to data collection tools and data collection procedures to improve clarity. More explanations have been added as well.

Comment 2 The manuscript still lacks a clear statement on data availability. 

 Author response: We have reviewed our statement; data has been made available as supplementary information. 

Comment 3 In terms of clarity, there are some grammatical inconsistencies, but nothing serious. 

 Author response: Thank you for your comment

Reviewer #3:

Comment 1 Abstract

The authors used an abbreviation of PE/E in the abstract you have to define it before you use it.

 Author response: thank you for your comment. The abbreviation PE/E has been omitted from the abstract. 

Comment 2 Result section

The authors are establishing thematic categories, like "Indications for magnesium sulfate use," solely relying on the perspective or opinion of a single individual. How did you ensure the validity of each thematic area, such as "Indications for magnesium sulfate use," when it's based solely on the perception or opinion of one individual? I recommend incorporating perspectives from other individuals regarding each thematic area. 

 Author response: Thank you for your feedback. We have revised the results section to incorporate viewpoints from various individuals.

Comment 3 Conclusion

Did the authors conclude based on participant reports or established facts that "Overall, magnesium sulfate is safe and beneficial for women and their babies when administered appropriately by nurse-midwives"? It would be preferable to conclude based on findings that align with the research objectives.

 Author response: We appreciate the comment. We have reworked the manuscript and made conclusion based on our findings.

---

## [Editor Report · Decision Letter 1]

23 Jul 2024

Perspectives of providing magnesium sulfate to patients with preeclampsia and eclampsia: A qualitative study amongst Nurse-midwives in Dar es Salaam, Tanzania

PONE-D-24-06856R1

Dear Mr Chikwala,

We’re pleased to inform you that your manuscript has been judged scientifically suitable for publication and will be formally accepted for publication once it meets all outstanding technical requirements.

Kind regards,

Ayodeji Babatunde Oginni, MSc

Academic Editor

PLOS ONE
---

## [Editor Report · Acceptance letter]

29 Jul 2024

PONE-D-24-06856R1 

PLOS ONE

Dear Dr. Chikwala, 

I'm pleased to inform you that your manuscript has been deemed suitable for publication in PLOS ONE. Congratulations! Your manuscript is now being handed over to our production team.

Kind regards, 

on behalf of

Dr Ayodeji Babatunde Oginni 

Academic Editor

PLOS ONE